# Approaches to Reducing Normal Tissue Radiation from Radiolabeled Antibodies

**DOI:** 10.3390/ph17040508

**Published:** 2024-04-16

**Authors:** Hiroyuki Suzuki, Kento Kannaka, Tomoya Uehara

**Affiliations:** Laboratory of Molecular Imaging and Radiotherapy, Graduate School of Pharmaceutical Sciences, Chiba University, 1-8-1 Inohana, Chuo-ku, Chiba 260-8675, Japan; kannaka@faculty.gs.chiba-u.jp (K.K.); tuehara@chiba-u.jp (T.U.)

**Keywords:** radioimmunotherapy, radioimmunodetection, radionuclides, PET, SPECT

## Abstract

Radiolabeled antibodies are powerful tools for both imaging and therapy in the field of nuclear medicine. Radiolabeling methods that do not release radionuclides from parent antibodies are essential for radiolabeling antibodies, and practical radiolabeling protocols that provide high in vivo stability have been established for many radionuclides, with a few exceptions. However, several limitations remain, including undesirable side effects on the biodistribution profiles of antibodies. This review summarizes the numerous efforts made to tackle this problem and the recent advances, mainly in preclinical studies. These include pretargeting approaches, engineered antibody fragments and constructs, the secondary injection of clearing agents, and the insertion of metabolizable linkages. Finally, we discuss the potential of these approaches and their prospects for further clinical application.

## 1. Introduction

Antibodies selectively bind to antigens with a high affinity and specificity and can be used as “magic bullets” [1,2]. Thus, antibodies are useful vehicles for delivering radionuclides to target tissues [3,4]. Radiolabeled antibodies for diagnosis or therapy can be prepared from parental antibodies by choosing radionuclides and are thus useful for radiotheranostics [5]. Radiolabeled antibodies have been widely developed for radioimmunodetection (RID) and radioimmunotherapy (RIT) so far [3,4,5]. Radionuclides for single-photon emission computed tomography (SPECT), such as indium-111 (^111^In), show suitable half-lives for the tracing of whole IgG and have often been used for RID [3,5]. The half-life of most radionuclides used in positron emission tomography (PET) is short, limiting the application of immuno-PET. However, the application of long-half-life PET radionuclides such as zirconium-89 (^89^Zr) has recently increased [4,6]. RIT is achieved by administering antibodies radiolabeled with beta or alpha emitters [7]. Although both radiolabeled antibodies have been preclinically and clinically investigated over the past three decades, clinical trials and approval for RIT using beta emitters have taken precedence [8]. Radiolabeled antibodies for RIT, [^90^Y]Y-ibritumomab tiuxetan (Zevalin) [9,10] and [^131^I]I-tositumomab (Bexxar) [11,12], were approved by the US Food and Drug Administration (FDA) in 2002 and 2003, respectively. Meanwhile, the approval of [^223^Ra]RaCl_2_ [13] and a report showing the high effectiveness of [^225^Ac]Ac-PSMA-617 [14] for the treatment of castration-resistant prostate cancer triggered the development of radiopharmaceuticals for targeted alpha therapy (TAT) [15,16]. Consequently, the number of clinical trials of RIT using alpha emitters has significantly increased over the past decade. There is no doubt that the further development of radiopharmaceuticals for RID and RIT will continue.

Radionuclides always emit radiation, and thus, avoiding radiation to normal tissues is very important, as well as the selective delivery of radionuclides to the targeted tissues. Because the dissociation of radionuclides from radiopharmaceuticals can cause side toxicities, radiopharmaceuticals must retain stable chemical bonds with radionuclides. Radiolabeling protocols have already been established for many radionuclides owing to numerous efforts to develop methods for radiolabeling antibodies. However, a few radionuclides still require improvement in radiolabeling methods. In particular, the redistribution of daughter radionuclides to normal tissues has been reported for alpha-emitters [17,18,19]. Other limitations need to be addressed for the clinical application of radiolabeled antibodies. The slow blood clearance of radiolabeled antibodies hinders the clear visualization of RID and causes bone marrow toxicity in RIT [20,21,22]. Accumulation in normal tissues, such as the liver and spleen, is problematic [23,24]. In addition, the heterogeneous accumulation of radiolabeled antibodies in tumor tissues is undesirable for accurate diagnosis and effective therapy. RIT has shown successful results in hematological cancers owing to the radiosensitivity and accessibility of radiolabeled antibodies to cancer cells [9,10,11,12]. However, the treatment of solid tumors has had limited success because of limited accessibility and a limited injection dose [8]. This review focuses on a strategy for reducing undesired radioactivity levels in the blood and normal tissues observed after the administration of radiolabeled antibodies. The mechanisms of blood and normal tissue accumulation have been introduced, and approaches for reducing these accumulations, mainly in preclinical studies, have been summarized.

## 2. Dissociation of Radionuclides

If radionuclides dissociate from radiopharmaceuticals, they are redistributed to normal tissues depending on the biodistribution properties of each element [25]. As a result, radiation is emitted in normal tissues, causing a decrease in diagnostic accuracy for imaging, a decrease in therapeutic efficacy, and an increase in the side effects of treatment. Therefore, radiolabeling methods that stably retain the chemical bonds between the radionuclides and antibodies are indispensable for the development of radiolabeled antibodies.

### 2.1. Dissociation of Radiohalogens

Although fluorine-18 (^18^F) is the most representative radionuclide used for PET, ^18^F has rarely been used to label whole IgGs because of its short half-life (110 min). The most typical and classical examples of radiolabeled IgGs are radioiodinated IgGs. Astatine-211 (^211^At)-labeled IgGs have long been investigated, and recent attention to TAT has resulted in an increasing number of reports of ^211^At-labeled IgGs. In addition, bromine-76 is applicable for PET. These radiohalogens share similar chemical properties, and radiolabeling methods have been used to produce radiohalogenated IgGs. However, the stability of radiohalogenated IgG and the metabolic fate of dehalogenated chemical species differ.

#### 2.1.1. Radioiodine and Radiobromine

When the first radioiodinated antibody was reported in 1953, ^131^I was used [26]. As ^131^I emits both gamma and beta-rays, ^131^I has been used in both RID and RIT. The half-life of ^131^I (8.02 d) was sufficiently long to match the slow biodistribution profiles of the antibodies. Although ^123^I also emits gamma-rays, its half-life (13.2 h) is relatively short for antibody tracing. ^124^I is one with a relatively longer half-life (4.2 d) and is useful for applications in immuno-PET. However, basic experiments on ^124^I-labeled PET probes are limited owing to their limited availability [27,28].

In the classical method, iodide (I^-^) is converted to the reactive electrophilic form (I^+^) by oxidants such as chloramine-T and iodogen [29] and reacts directly with proteins (Figure 1A). The electrophilic substitution of I^+^ occurs at the ortho-position of the phenolic hydroxyl group of tyrosine and, to some extent, on the imidazole nitrogen of histidine. This method is called direct radioiodination and can be performed using a simple procedure that usually yields good products. Unfortunately, proteins radioiodinated via direct radioiodination are easily metabolized by dehalogenation enzymes to release I^-^, which accumulates in tissues highly expressing sodium/iodide symporters (NIS), such as the stomach and thyroid. To solve this problem, an indirect radioiodination method was developed. Because dehalogenation enzymes often catalyze the formation of an ortho-iodophenol moiety, iodophenyl moieties are used indirectly. Radioiodine is introduced at the meta- or para-position of the phenyl group, and radioidinated proteins with high in vivo stability can be obtained in both cases. The agents for indirect radioiodination, *N*-succinimidyl 3- or 4-iodobenzoate (SIB), were reported by Zalutsky et al. [30] and Wilbur et al. [31]. SIB can be easily prepared from a trialkylstannyl precursor and used for radiolabeling proteins using a well-established protocol [32] (Figure 1B). Thus, this method has widely been used to prepare radioiodinated proteins.

Debromination also occurs in radiobrominated proteins using direct methods [33]. Radiobromine shows nonspecific accumulation in the extracellular space [34], resulting in persistent localization to the blood and organs [33]. Indirect radiobromination is also helpful for reducing the dehalogenation of radiobrominated and radioiodinated compounds (Figure 1B). Since radiobromination requires stronger and larger amounts of oxidants than radioiodination [35], indirect radiobromination is also beneficial for retaining the inherent affinity of the parental antibodies. The reaction conditions for *N*-succinimidyl 4-bromobenzoate were optimized by Höglund et al. [36].

#### 2.1.2. Astatine

^211^At is one of the most promising alpha-particle emitters, and the preparation of ^211^At-labeled proteins has been investigated for over four decades [37]. Astatine is not naturally occurring, as it has no stable isotopes, and its half-life of the longest one (^210^At) is only 8.1 h. The decay of ^210^At produces ^210^Po, which is highly radiotoxic and was notoriously used for the assassination of former Russian intelligence member Alexander Litvinenko [38]. Hence, ^210^At cannot be applied to TAT. The isotope ^211^At is alpha-emitting, with a half-life of 7.2 h. Its lack of general availability has, however, limited ^211^At research. General chemical analytical techniques cannot be used for ^211^At-labeled compounds; they are often characterized by their corresponding iodinated compounds. Although the chemical properties of the astatinated and iodinated compounds are sufficiently similar for the characterization of astatinated compounds, there are some differences. The most problematic difference is the in vivo stability. As described above, indirect radioiodination using SIB provides radioiodinated proteins with high in vivo stability, but astatinated proteins radiolabeled with *N*-succinimidyl 3- or 4-astatobenzoate (SAB) cause in vivo deastatination [39,40] (Figure 1B). When free astatine is released, it accumulates in the thyroid and stomach, similar to iodine [39]. Free astatine accumulates in the lungs and spleen. Astatine is more easily oxidized than iodine, and the oxidation of At^-^ to At^+^ may cause its accumulation in the lungs and spleen. This hypothesis was supported by the finding that the pre-administration of periodate before [^211^At]At^-^ administration increased its accumulation in these tissues [41].

Garg et al. and Hadley et al. compared the biodistribution of astatinated IgGs and Fabs and observed higher radioactivity levels in the stomach and thyroid for ^211^At-labeled Fabs than for IgGs at an early post-injection time [39,40]. In addition, the administration of ^211^At-labeled biotin derivatives in mice resulted in high accumulation in these organs early after injection [42]. These findings imply that in vivo deastination occurs more rapidly when lower-molecular-weight compounds are labeled with ^211^At. Low-molecular-weight compounds accumulate more rapidly in each tissue from the bloodstream and are metabolized, and in vivo deastatination probably occurs after degradation in the cells. Carbon-astatine bonds are well known to show lower stability than carbon-iodine bonds [43], which have been considered to cause in vivo deastatination. This contradicts the different stabilities of ^211^At-labeled IgGs and Fabs. Teze et al. proposed that Fenton and Fenton-like reactions in lysosomes may be involved in in vivo deastatination [44]. These reactions produce oxidative forms of ^211^At-labeled compounds, which further impair the bond dissociation energies of C-At bonds and trigger in vivo deastatination in the lysosomes. They also proposed that cytochrome-P450s (CYPs) may be involved in oxidative deastination.

Although the slow targeting property of antibodies is not considered to match the relatively short half-life of ^211^At, research on ^211^At-labeled antibodies has been performed partly because of the limitation in their application to low-molecular-weight compounds. Thus far, the enhancement of stability has been investigated. Wilbur et al. focused on the fact that the B-At bond has a higher bond dissociation energy than that of the C-At bond and developed boron cage moieties (Figure 2A). They developed ^211^At-labeled Fab’ fragments using boron-cage moieties to demonstrate they showed high stability against in vivo deastatination [45]. However, boron cage moieties have prolonged retention in normal tissues, which limits their applicability [46]. Meanwhile, the development of ^211^At-labeled compounds containing C-At bonds is ongoing. In the last decade, several stable 2^11^At-labeled compounds have been developed, including low-molecular-weight antibody constructs, peptides, and middle- and low-molecular-weight compounds [47,48,49,50]. Among these, the guanidinomethyl benzoyl moieties have the potential to serve as versatile scaffolds for the preparation of ^211^At-labeled antibodies [47] (Figure 2B). The aryl C-At bond has been considered the only choice for the sufficient chemical stability of the C-At bond against in vivo deastatination [16], and all of them use aryl C-At bonds [47,48,49,50]. Our recent studies produced stable 2^11^At-labeled low-molecular-weight compounds using alkyl C-At bonds [51] (Figure 2C), implying that in vivo deastatination may be caused by biological instability rather than chemical instability. Further studies investigating novel ^211^At-labeling systems that are not chemically stable may help to elucidate in vivo deastatination mechanisms. Although the stability of ^211^At-labeled IgGs against in vivo deastatination is compromised, they clearly show in vivo deastatination compared with radioiodinated IgGs [52]. The elucidation of the in vivo deastatination mechanisms would be helpful in developing ^211^At-labeled IgGs with a higher stability against in vivo deastatination.

### 2.2. Dissociation of Radiometals

Because radiometals cannot form stable chemical bonds directly with a targeting molecule, bifunctional chelating agents (BFCs), including chelating and conjugation parts with a targeting molecule, are used to label radiometals [53,54]. After the administration of radiometal-labeled agents, they become extremely diluted (over 2000 times) in living systems. Therefore, kinetic rather than thermodynamic stability is important for radiometal-chelator complexes. In addition, blood contains proteins that can bind strongly to metals, such as transferrin, albumin, and ceruloplasmin [53,55,56]. Concentrations of these proteins in the blood are relatively high and thus much higher than those of radiometal-labeled agents. If radiometals dissociate from the chelator, they can no longer reproduce the original radiometal-labeled agents via recomplexation. Therefore, a suitable choice for each radiometal is critical to avoiding the in vivo dissociation of radiometals. Combinations of radiometals and BFCs providing stable complexes have been described previously [25,57,58]. In this review, representative methods for producing radiolabeled antibodies have been described.

#### 2.2.1. Representative Choices of the BFCs for Preparing Radiometal-Labeled Antibodies

Classical chelating agents, EDTA and DTPA (Figure 3), are versatile and applicable to various radiometals, but they show low or moderate degrees of dissociation [59]. Macrocyclic chelators such as DOTA (Figure 3) show more kinetically inert complexes than those formed by acyclic chelators such as EDTA and DTPA [60], which is known as the macrocyclic effect [61]. Among several bifunctional DOTA derivatives, *p*-SCN-Bn-DOTA (Figure 3) is the most representative because it is commercially available, easy to handle, and can be conjugated using a well-established protocol [62]. Thus, *p*-SCN-Bn-DOTA is the most representative BFC for the development of radiopharmaceuticals with various radiometals, such as ^111^In, ^90^Y, ^177^Lu, ^225^Ac, ^212/213^Bi, ^227^Th, and ^212^Pb [25]. However, DOTA requires elevated conditions for radiolabeling. DOTA is now used as the current gold standard chelator for several radiometals in the radiolabeling of peptides [63,64], but other choices still have been searched for the radiolabeling of proteins. Several DTPA derivatives, such as 1B4M-DTPA and CHX-A’’-DTPA (Figure 3), have been developed to enhance the stability of DTPA complexes with various radiometals [65,66]. They formed sufficiently stable In/Y-complexes; indeed, 1B4M-DTPA was used as the BFC in Zevalin. Other acyclic chelators have also been developed [67,68,69].

#### 2.2.2. Preparation of ^89^Zr-Labeled Antibodies

The representative radiometal applicable to immuno-PET is zirconium-89 (^89^Zr) due to its long half-life (3.3 d), sufficient to achieving optimal tumor-to-blood ratios before decay. ^89^Zr decays via positron emission (23%) and electron capture (77%). Desferrioxamine (DFO) is the most representative chelator for ^89^Zr-labeling and has been used in clinical applications [70]. However, DFO cannot satisfy the octadentate coordination geometry of Zr [71] and lacks the in vivo stability of a complex with ^89^Zr [72]. Free ^89^Zr accumulates in the bone [73], and the instability of the [^89^Zr]Zr-DFO complex was observed as a high uptake of ^89^Zr in the bone [74]. The bone accumulation of ^89^Zr is a major drawback of immuno-PET using ^89^Zr. The dissociation of ^89^Zr increases with time, and the slow blood clearance of antibodies and the long half-life of ^89^Zr result in a substantial radiation dose to the bone. Novel BFCs for ^89^Zr are being developed to address this issue [75], but DFO is still mainly used.

#### 2.2.3. Preparation of ^64^Cu-Labeled Antibodies

Another suitable choice for immuno-PET using whole IgG is copper-64 (^64^Cu)-labeled antibodies. ^64^Cu decays via positron emission (18%), β^-^ (38%) and electron capture (44%) with 12.7 h of half-life. Free copper ions are incorporated by human copper transporter 1 (hCtr1), which is highly expressed in the liver [76]. Thus, the dissociation of ^64^Cu from the parental ^64^Cu-labeled antibodies causes the accumulation of ^64^Cu in the liver. In addition, ^64^Cu-chelator complexes with insufficient stability may cause the transchelation of proteins such as ceruloplasmin [77,78,79]. Ceruloplasmin is a protein produced in the liver that is involved in the transport and metabolism of copper. Approximately 70% of the Cu in plasma binds to ceruloplasmin, and the rest binds to albumin (15%) or macroglobulins [80]. Another transchelation pathway was observed when ^64^Cu-labeled octreotide analogs were administered to normal rats. Approximately 70% of ^64^Cu is transchelated from the parental peptides to proteins with a size of 35 kDa, which are considered superoxide dismutases (SOD) [81]. Among the three oxidation states of Cu(I–III), Cu(II) is considered the best suited for the radiolabeling of antibodies [82]. Notably, the reduction of Cu(II) to Cu(I) can also trigger the loss of ^64^Cu from the parental ^64^Cu-labeled antibodies. Therefore, resistance to the reduction of Cu(II) to Cu(I) is required for ^64^Cu-chelator complexes as well as kinetic inertness. Because of its versatility over other radiometals, DOTA has often been used to prepare ^64^Cu-labeled antibodies [82,83,84,85]. However, DOTA is not sufficiently stable. Although TETA provides a more stable ^64^Cu-complex than DOTA [86], [^64^Cu]Cu-TETA-labeled octreotide causes the transchelation of SOD in the liver, as described above [81]. NOTA, another classical chelator, formed a relatively stable complex with ^64^Cu [87]. Promising BFCs for ^64^Cu have been developed, including cross-bridged azamacrocyclic chelators [88,89,90], sarcophagine derivatives [91], and bispidine derivatives [92].

#### 2.2.4. Preparation of ^225^Ac-Labeled Antibodies

^225^Ac emits four alpha particles per atom of ^225^Ac with a half-life of 10 d. Initially, DOTA was used to radiolabel antibodies with ^225^Ac, but its radiolabeling efficiency was low [93]. Unfortunately, the ^225^Ac-labeled antibody using CHX-A’’-DTPA as a BFC showed low stability in releasing a free ^225^Ac ion [94]. Thus, the DOTA radiolabeling procedure was optimized for ^225^Ac to increase the yield [94]. Recently, Macropa has gained attention because it exhibits excellent chelating properties suitable for ^225^Ac [95]. When Ac dissociates from a chelator, it mainly accumulates in the liver [95]. However, there are good choices for BFCs that provide excellent in vivo stability to ^225^Ac-complexes, and the in vivo dissociation of ^225^Ac can almost be ignored.

### 2.3. Dissociation of Daughter Radionuclides

Many alpha-emitting radionuclides have a problem in that their daughter radionuclides dissociate from BFCs. Although the direct in vivo dissociation of ^225^Ac from the chelator is unlikely, the dissociation of daughter radionuclides is regarded as an unavoidable phenomenon [18]. Generally, daughter nuclides receive recoil energy upon alpha decay. The recoil energy was at least 100 keV, which is more than 1000 times higher than the binding energy of any chemical bond. Therefore, the dissociation of daughter radionuclides is unavoidable even when excellent BFCs are used. In the case of ^225^Ac, the longest half-lived daughter nuclide is ^213^Bi (45.6 min), which accumulates in the kidney [96]. ^221^Fr is another radionuclide with a relatively long half-life (4.9 min) among alpha-emitting daughter radionuclides of ^225^Ac. Francium is an alkali metal reabsorbed by renal tubular cells. Hence, renal accumulation is the major cause of radiation damage from daughter radionuclides of ^225^Ac. Another example of an alpha-emitting radionuclide that shows the dissociation of daughter radionuclides is ^227^Th. The decay of ^227^Th produces ^223^Ra, which is liberated from BFCs by the recoil, and radium is well known to accumulate in the bone.

^212^Pb produces alpha particles that emit ^212^Bi by beta decay, and its complex stability (bond energy: ~10 eV) against the recoil energy of beta rays (0.5 eV) seems to be tolerable. However, the dissociation of ^212^Bi from DOTA has also been reported [97]. Because the dissociation ratio corresponds to that of the internal conversion, ^212^Bi was assumed to dissociate from the chelator originating from the excitation of electrons during the internal conversion process.

#### 2.3.1. Administration of Agents to Prevent Renal Toxicity

Because dithiol chelators form Bi-complexes with high urinary excretion, Jaggi et al. investigated the administration of dithiol chelators before treatment with ^225^Ac-labeled antibodies [19]. Two dithiol chelators, 2,3-dimercapto-1-propanesulfonic acid (DMPS) and *meso*-2,3-dimercaptosuccinic acid (DMSA), were investigated; the former was found to be more effective in reducing the renal accumulation of ^213^Bi. Jaggi et al. investigated the administration of furosemide and thiazide diuretics, which are known to inhibit the renal reabsorption of alkali metals [19]. They comparably reduced the renal accumulation of both ^211^Fr and ^213^Bi, and the combination of DMPS caused a dramatic reduction in ^213^Bi accumulation in the kidney (75–80%). Jaggi et al. reported that spironolactone was effective in preventing nephrotoxicity caused by alpha-particle radiation from daughter radionuclides [98]. Hence, furosemide and spironolactone were administered to patients in clinical trials of ^225^Ac-labeled antibodies [99]. In addition, this strategy was applied to ^212^Pb-labeled antibodies because their alpha particle-emitting daughter radionuclide is ^212^Bi [100].

#### 2.3.2. Strategies for Minimizing the Side Toxicity Caused by Daughter Radionuclides

In a review by Kruijff, three strategies for dealing with the dissociation of daughter radionuclides from alpha radionuclides were introduced and discussed [18]. One is encapsulation in nanocarriers. The second approach involves the use of targeted molecules that show a rapid uptake by tumor cells. After ^225^Ac is internalized by tumor cells, the liberated ^221^Fr and ^213^Bi ions are retained in the tumor cells [101]. Therefore, the rapid delivery of ^225^Ac to targeted tumor tissues followed by rapid internalization is a reasonable approach. In this case, all alpha-ray emissions from ^225^Ac and daughter radionuclides occurred in the tumor cells, which is useful for highly effective treatment. However, antibodies show slow tumor accumulation, and low-molecular-weight targeting molecules are more suitable for this approach. The third approach is the local administration of radiolabeled agents. Although radiolabeled antibodies can be used in this approach, their clinical application is limited. Therefore, a novel strategy for enhancing the safety of RIT using alpha-emitting radionuclides is required.

## 3. Slow Blood Clearance

Prolonged blood retention is a major drawback of radiolabeled antibodies. A high tumor/blood ratio is desirable for clear imaging, and the reduction in radioactivity levels in the blood and high tumor accumulation are important. High blood radioactivity levels result in myelotoxicity in RIT owing to the high radiation sensitivity of the bone marrow. Indeed, myelotoxicity is generally the major side effect of radiolabeled antibodies. Slow blood clearance also results in slow tumor accumulation, which limits the application of short-half-lived radionuclides. In this review, three strategies for reducing the prolonged blood retention of radiolabeled antibodies are introduced: pretargeting approaches, the use of antibody fragments and constructs, and the secondary injection of clearing agents (Figure 4).

### 3.1. Pretargeting System

The pretargeting system is a strategy for avoiding the problems associated with the prolonged circulation time of radiolabeled antibodies (Figure 4A). In this system, antibodies with tags (not radiolabeled) are first administered. After sufficient time has elapsed for eliminating these antibodies from the blood pool, a radiolabeled secondary agent that binds to the tag is administered. RID and RIT with very low blood radioactivity levels can be achieved in this system because radionuclides can be delivered independent of the slow accumulation of antibodies. Hence, low-molecular-weight compounds are generally used for the secondary agents. In vivo conjugation occurs on the membrane of the cancer cells, and antibodies are desired to not be internalized. Extremely effective and selective conjugation is required for this approach, and four pretargeting systems have been developed [102].

#### 3.1.1. Biotin-Avidin System

The affinity between biotin and avidin is one of the strongest noncovalent interactions in nature. Biotin is a 244 Da vitamin found in low concentrations in the blood and tissues. Avidin and streptavidin are 65 and 56 kDa proteins, respectively, and are composed of four subunits. Each subunit can bind to a single biotin molecule with a dissociation constant of 10^−15^ M. Streptavidin is more suitable for the pretargeting approach because it shows less nonspecific binding to normal tissues than avidin. The application of the biotin-avidin system to the pretargeting approach was first reported in 1987 by Hnatowich et al. [103,104]. Because radiolabeled streptavidin shows relatively slow blood clearance, radiolabeled biotin derivatives are generally administered for conjugation with pretargeted antibodies via streptavidin [105]. The urea moiety and the non-oxidized thioether structures in biotin are responsible for binding to avidin; thus, the carboxylic acid was initially used for conjugation with radiolabeling moieties. Hence, the simplest strategy for preparing radiolabeled biotin derivatives is direct conjugation with the carboxyl group of biotin to form a CO-NH amido bond [106]. However, this conjugation method causes in vivo instability against metabolism by biotinamide amino hydrolases (EC 3.5.1.12, biotinidase) [107], and two strategies have been developed to avoid biotinidase degradation. One is to increase the steric hindrance of the neighboring amide bond [108], and the other is to change the CO-NH amido bond to NH-CO [109].

Streptavidin-conjugated antibodies are slowly cleared from blood circulation, which interferes with the binding of radiolabeled biotin administered at intervals after preloading [105]. To solve this problem, a three-step method is proposed [110]. In this method, avidin/streptavidin is administered as a clearing agent to remove the circulating biotinylated antibodies. Finally, radiolabeled biotin is administered to bind the preloaded antibody to the tumor via avidin/streptavidin. Owing to these improvements in preclinical studies, the biotin-avidin system has been applied in several clinical trials [111]. However, these trials revealed a severe drawback of the biotin-avidin system. After the first pretargeting regimen, the formation of human anti-streptavidin antibodies was observed, which could cause allergenic responses during the second or third course of the pretargeting regimen. Another problem is the existence of endogenous biotin in the blood, which causes saturation of the biotin conjugation sites of streptavidin conjugated with antibodies before the administration of radiolabeled biotin derivatives.

#### 3.1.2. Bispecific Antibody

Another classical pretargeting system uses a bispecific antibody containing two different Fab fragments. One is for conjugation with antigens expressed on target cells and the other is for conjugation with radiolabeled haptens. Initially, a bispecific antibody targeting chelating agents, such as [^111^In]In-EDTA, was used as the pretargeting system [112]. This system has a major drawback in its applicability to other radiometals; changing the radiometals critically impairs the binding affinity to the bispecific antibody [113]. To tackle this problem, a system using an anti-hapten antibody that binds to histamine succinyl glycine (HSG) was developed [114]. Because the binding affinity of this system is independent of the radiometals conjugated with the chelators, the applicability of the pretargeting system using bispecific antibodies was expanded. The initial pretargeting system of bispecific antibodies had another problem in the reduction in the tumor localization of radioactivity due to the univalent interaction with targeting antigens and radiolabeled haptens. The former results in a low accumulation of bispecific antibodies in tumors, and the latter results in a low retention of radioactivity owing to the dissociation of radiolabeled haptens. Radiolabeled bivalent haptens [115,116] and tri-Fab bispecific antibodies that include two Fabs against tumor-associated antigens [117] have been developed.

Several clinical trials of pretargeting systems using bispecific antibodies have been performed for both RIT and RID. Clinical optimization studies of the ^177^Lu-labeled hapten peptide ([^177^Lu]Lu-IMP288) and tri-Fab bispecific antibody (TF2) were reported in 2013 by Schoffelen et al. [118]. In this study, different dose schedules in patients with progressive metastatic colorectal cancer were used to investigate the interval between TF2 and [^177^Lu]Lu-IMP288 administration, the dose escalation of TF2, and the dose reduction of peptides. Tumor accumulation was improved by changing intervals from 5 days to 1 d and the peptide doses from 100 to 25 µg. The high radioactivity of [^177^Lu]Lu-IMP288 (2.5–7.4 GBq) was well tolerated, and feasibility and safety were demonstrated. Another optimization study of a similar pretargeting system was performed by Bodet-Milin et al. [119], and molar doses and pretargeting intervals between TF2 and ^68^Ga-labeled IMP288 for immuno-PET in relapsed medullary thyroid carcinoma patients were investigated. The results demonstrated that high-contrast tumor uptake could be obtained, especially using optimized parameters such as a bispecific antibody-to-peptide mole ratio of 20 and an interval of 30 h.

Many bispecific antibodies have also been developed as anticancer agents, and their clinical applications have revealed their immunogenicity caused by bispecific antibodies [120]. Bodet-Milin et al. reported that premedication with corticosteroids and antihistamines before the injection of bispecific antibodies could induce transient immunosuppression, limiting immediate and delayed immune effects [119]. Indeed, this premedication reduced the immunization rate compared with studies using the same compounds without premedication (16% versus 52%) [118]. The effectiveness of this approach is supported by previous studies [121,122].

#### 3.1.3. Oligonucleotide

The third pretargeting system uses an oligomer and a complementary oligomer. Because native DNA and RNA are unstable against degradation by nucleases, chemical analogs of single-stranded DNA have been developed to improve metabolic stability. These include peptide nucleic acids (PNA) [123], morpholino oligomers (MORFs) [124], and mirror-imaged oligonucleotides [125]. These systems have provided promising results in preclinical studies. However, to the best of our knowledge, no clinical trials have yet been conducted.

#### 3.1.4. Bioorthogonal Chemistry

Bioorthogonal chemistry enables high-yield, rapid, and selective chemical reactions in biological environments and shows no reactivity towards endogenous functional groups. Although several bioorthogonal reactions such as Staudinger ligation and strain-promoted azide–alkyne cycloaddition have been developed [126], they are not suitable for pretargeting applications because of their slow reaction rates. This drawback was overcome by the development of an inverse electron-demand Diels-Alder (iEDDA) reaction. Blackman et al. reported that a reaction between 3,6-di-(2-pyridyl)-*s*-tetrazine and *trans*-cyclooctene (TCO) exhibited a very rapid reaction rate (*k*_2_ 2000 M^−1^ s^−1^) [127]. Subsequently, Rossin et al. applied this bioorthogonal reaction to a pretargeting system in which a TCO-conjugated antibody and ^111^In-labeled tetrazine were used [128]. The stability of ^111^In-labeled tetrazine was not high but sufficient considering its low half-life in blood (9.8 min). In addition, the TCO tag conjugated to the antibody showed in vivo deactivation (25% in 24 h). Therefore, the interval between the antibody and tetrazine probe administration was set to 24 h. According to the calculations in this report, the reaction yields in the tissues were 57% in the blood and 52% in the tumors, demonstrating that the iEDDA reaction is a promising pretargeting system. In a subsequent study, Rossin et al. investigated the deactivation mechanism of TCO [129]. This study revealed that the conversion of TCO to the *cis*-isomer, which is five orders of magnitude less reactive toward tetrazines than TCO, is responsible for the in vivo deactivation of TCO. They proposed that the redox interaction of TCO with protein-bound Cu(II) caused trans-to-cis conversion. Based on this consideration, they removed the PEG linker inserted between the TCO and the antibody to increase the bulkiness surrounding the TCO, which resulted in the improved stability of the TCO without impairing the reaction rates. Finally, they performed an SPECT/CT imaging experiment in which a 3-day interval was set. As a result, background levels decreased at the 3-day interval compared to that at the 1-day interval. Notably, in vivo deactivation was still observed, although it was relatively slow. In conclusion, the stability of the tag molecules may not be problematic, and promising results have been reported so far [130].

### 3.2. Antibody Fragments and Constructs

Radiolabeled antibody fragments and constructs show more rapid blood clearance and distribution in tumor tissues than intact IgG antibodies (Figure 4B). In addition, they show an evener distribution than radiolabeled antibodies, which is an advantage that cannot be obtained using pretargeting approaches. However, the advantages of these antibodies are limited owing to their decreased accumulation in tumors [131]. In addition, radiolabeled antibodies exhibit high levels of radioactivity in the kidney, which is the main dose-limiting organs [132]. Renal accumulation also impairs the precise imaging proximal to the kidney, but more critically, it impairs the therapeutic effects of RIT. Both the injection dose and the tumor accumulation after injection are limited by renal accumulation and impaired binding affinity, respectively, resulting in impaired therapeutic effects.

Because the membrane surface of the proximal tubule is negatively charged, the co-administration of positively charged amino acids is effective in competitively inhibiting the reuptake of radiolabeled antibody fragments and constructs via electrostatic interactions [133,134]. However, this strategy can cause side effects, such as nausea, vomiting, and hyperkalemia. Another approach is the renal brush border strategy, which introduces a cleavable peptide linkage between antibody fragments and the radiolabelled part [135]. This strategy was initially developed as a radioiodinated antibody Fab fragment in 1999 [136] but was difficult to apply to radiometals, with the exception of ^188^Re-labeled Fab [137]. In the past decade, we succeeded in applying the renal brush border strategy to several radiometals, such as ^99m^Tc, ^67^Ga, ^64^Cu, and ^111^In [138,139,140,141,142]. Although this strategy has not yet been applied to RIT, the dramatic reduction in renal radioactivity may contribute to enhanced therapeutic effects.

### 3.3. Clearing Agents

As used in the three-step pretargeting method for the biotin-avidin system, avidin can be used as a clearing agent to remove biotinylated antibodies from the blood circulation (Figure 4C). Yudistiro et al. proposed a strategy that involved the initial administration of the ^90^Y-labeled biotinylated antibody and subsequent avidin administration at an interval [143]. Indeed, the ^90^Y-labeled biotinylated antibody was rapidly cleared from the blood 3 h after the administration of avidin with a 24 h interval from antibody injection. In the present study, tumor accumulation was not impaired, demonstrating the effectiveness of this strategy. In contrast, the ^90^Y-labeled biotinylated antibody accumulates in the liver and exhibits the persistent localization of radioactivity, which may cause liver toxicity. As pointed out by Yudistiro et al., the strategy requires further optimization, such as the dose and timing of the avidin injection, when applied in clinical practice [143]. However, the results indicated the potential usefulness of this strategy.

## 4. Accumulations in the Liver and Spleen

Radiolabeled antibodies exhibit a high accumulation of radioactivity in tumor tissues, as well as in the liver and spleen [24]. The sensitivity of RID depends on the contrast in radioactivity between tumors and their surrounding tissues; however, metastasis to the liver is observed in many cancers such as gastrointestinal, breast, and ovarian cancers [144]. Therefore, a reduction in accumulation in the liver and spleen is important for a more accurate RID. Proteins that are not filtered by the glomerulus generally accumulate in the reticuloendothelial system, including the liver and spleen, and are then catabolized in the lysosomes, although antibodies are partially recycled into blood circulation via conjugation with neonatal Fc receptors. When radiolabeled antibodies accumulate in the liver and spleen, lysosomal degradation produces the final radiometabolites, which are responsible for the localization of radioactivity in the liver and spleen [24]. The persistent localization of radioactivity in the liver and spleen is predominant for radiometal-labeled antibodies, because the final radiometabolites contain radiometal–chelator complexes that are generally highly hydrophilic and bulky. The choice of chelator can sometimes contribute to decreasing radioactivity levels in the liver and spleen. For instance, ^111^In-labeled proteins that produced [^111^In]In-EDTA-lysine as the final radiometabolite showed a shorter residence time in the liver than those producing [^111^In]In-DTPA-lysine and [^111^In]In-DOTA-lysine [145,146]. Although the elimination rate of [^111^In]In-EDTA-lysine from the liver was much lower than that of radioiodinated proteins, the results suggest the usefulness of the strategy consisting of the rapid and selective liberation of the radiometabolite and the subsequent rapid elimination of the radiometabolite from the liver lysosomes.

### 4.1. Use of an Ester Bond as the Cleavable Linkage

Hippuric acid, a conjugation product of benzoic acid and glycine, is a representative compound that undergoes rapid urinary excretion from the liver. Therefore, [^125/131^I]*m*-iodohippuric acid is a promising candidate radiometabolite for reducing radioactivity levels in the liver and spleen. A strategy for producing [^125^I]*m*-iodohippuric acid as a radiometabolite was initially designed by introducing an ester bond as a cleavable linkage (Figure 5) [147]. A reagent containing an ester bond (maleimidoethyl *m*-iodohippurate (MIH)) was used to radiolabel intact IgG and Fab fragments against osteogenic sarcoma (OST7) to produce MIH-OST7 and MIH-Fab. While [^125^I]MIH-IgG reduced hepatic radioactivity levels, it showed more rapid blood clearance and lower tumor accumulation at 48 h post-injection than radioiodinated antibodies via the SIB method. These results suggested that the ester bond was cleaved in the blood to accelerate blood clearance. Impaired tumor accumulation is caused by accelerated blood clearance and/or cleavage of the ester bond in tumor tissues. In addition, [^125^I]MIH-Fab exhibited rapid ester bond cleavage in both in vitro and in vivo studies, implying that this strategy is not applicable to lower-molecular-weight proteins. To overcome these limitations, we developed a strategy for liberating radiometabolites with improved plasma stability.

### 4.2. Use of a Peptide Linkage as the Cleavable Linkage

To improve the stability of ester bonds, a peptide linkage cleaved by lysosomal degradation has been proposed [148]. Based on the strategy, 3′-iodohippuryl *N*^ε^-maleoyl-L-lysine (HML) was developed (Figure 5), in which the Gly-Lys sequence was inserted between the benzoyl moiety as a radiolabeling part and the maleimide group as a conjugation part with an antibody. HML releases the same radiometabolite, [^125^I]*m*-iodohippuric acid, as MIH in liver lysosomes. In contrast to the plasma instability of the ester bond, the Gly-Lys linkage showed significantly improved stability. When incubated in human serum, [^125^I]MIH-IgG released more than 50% of its initial radioactivity after 24 h, whereas [^131^I]HML-IgG released less than 1% of its radioactivity during the same interval. In biodistribution studies, [^131^I]HML-IgG showed similar radioactivity levels in the tumor as [^125^I]MIH-IgG at 24 h post-injection but was significantly higher at 48 h post-injection. In addition, [^131^I]HML-IgG showed radioactivity levels in the tumor and blood comparable to those of [^111^In]In-EDTA-labeled IgG 24 and 48 h post-injection. Simultaneously, [^131^I]HML-IgG exhibited a significantly higher tumor-to-liver ratio than [^111^In]In-EDTA-labeled IgG. These results indicate that [^131^I]HML-IgG can reduce radioactivity levels in the liver without impairing tumor accumulation.

However, when this strategy was applied to radiometal-labeled antibodies, radioactivity levels in the liver were not effectively reduced compared to radioiodinated antibodies [149]. The control of releasing the designed radiometabolites in lysosomes is relatively easy. However, it is difficult to design radiometabolites that can rapidly escape from lysosomes owing to the high hydrophilicity and bulkiness of the radiometal-chelator complex. As a result, the liberated radiometabolites are often retained in the lysosomes even if the metabolizable linkage is cleaved as expected. Our continuous efforts to develop radiolabeled antibody fragments with low renal radioactivity levels suggested that the radiometabolites used for the renal brush border strategy could have a short residence time in lysosomes [139]. Based on this finding, we recently applied a molecular design to reduce the renal radioactivity levels of ^111^In-labeled antibody fragments to ^111^In-labeled antibodies (Figure 5) [150]. As expected, the radiometabolite [^111^In]In-DO3A*i*Bu-Bn-F, which was rapidly excreted into the urine after liberation by renal brush border membrane enzymes, showed rapid clearance from the liver after the liberation of the parent antibody in the hepatic lysosomes. As a result, a higher tumor-to-liver ratio was achieved using this strategy compared to antibodies radiolabeled with ^111^In using the conventional method. The results suggest that a molecular design that liberates radiometabolites by the cleavage of metabolizable peptide linkages would be useful for reducing high radioactivity levels in the liver for radiolabeled antibodies, as well as in the kidney for radiolabeled low-molecular-weight antibody fragments and constructs.

## 5. Conclusions

Although there are unresolved problems, radiolabeling techniques for antibodies have grown and matured owing to numerous efforts to develop radiolabeled antibodies. Radiohalogenation protocols for antibodies have been well established, with the exception of ^211^At. Recently, several stable ^211^At-labeled compounds that may provide approaches to enhancing the in vivo stability of ^211^At-labeled antibodies have been reported. Most radiometals can be stably conjugated with the targeting molecules by choosing suitable BFCs, with a few exceptions, such as ^89^Zr. Meanwhile, the recently focused attention on TAT revealed that the dissociation of daughter radionuclides by recoil energy could be a limitation in the clinical application of TAT. Therefore, the development of novel strategies for overcoming this problem is highly desired.

Recent advances in the development of strategies for improving the effectiveness of radiolabeled antibodies for RIT and RID have provided promising results and hope for clinical applications. Representative approaches to reducing radioactivity levels in blood include the use of a pretargeting system and antibody fragments. Although the major drawback of immunogenicity limits the clinical application of a pretargeting approach using the biotin-avidin system, other systems, such as those utilizing bispecific antibodies, oligonucleotides, and bioorthogonal chemistry, have shown potential as alternative systems for pretargeting. Although antibody fragments have a major drawback causing high and persistent radioactivity levels in the kidney, some solutions have been proposed. Another strategy involving the administration of clearing agents is also useful for reducing radioactivity levels in the blood. In addition, an approach to reducing hepatic radioactivity levels is useful, particularly for RID. These approaches have not yet been sufficiently investigated for clinical application, and the results provide a basis for future applications of various kinds of RID and RIT.

## Figures and Tables

**Figure 1 pharmaceuticals-17-00508-f001:**
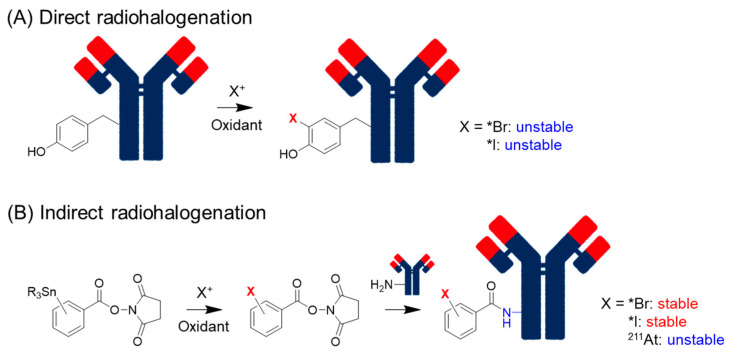
Direct and indirect radiohalogenation methods.

**Figure 2 pharmaceuticals-17-00508-f002:**
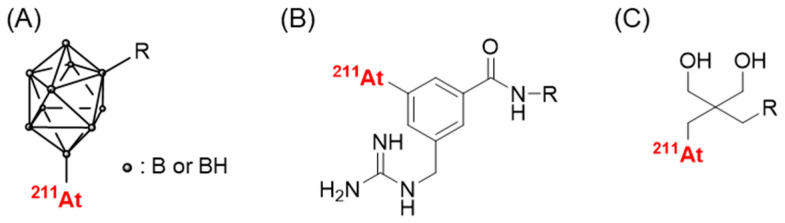
Scaffolds providing ^211^At-labeled compounds with high in vivo stability. (**A**) *closo*-decaborate(2-), (**B**) 3-[^211^At]astato-5-guanidinomethyl benzoyl, and (**C**) neopentyl glycol scaffolds.

**Figure 3 pharmaceuticals-17-00508-f003:**
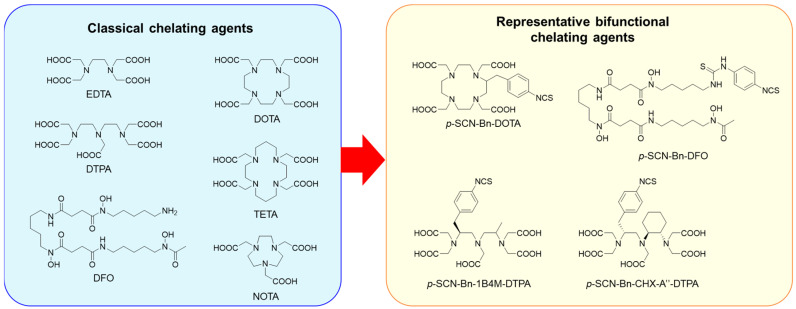
Classical chelating agents and representative BFCs for radiometals.

**Figure 4 pharmaceuticals-17-00508-f004:**
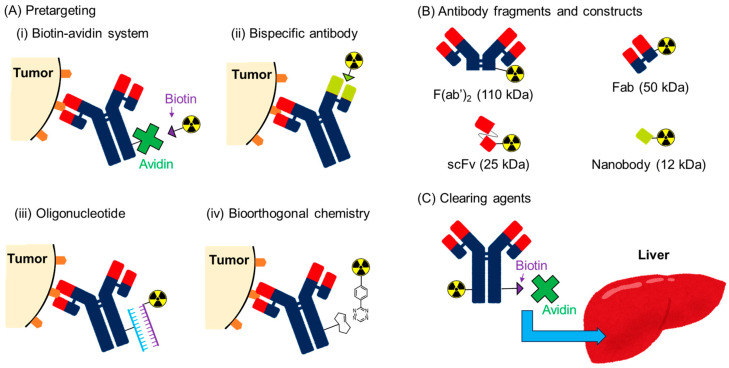
Strategies for reducing the prolonged blood retention of radiolabeled antibodies.

**Figure 5 pharmaceuticals-17-00508-f005:**
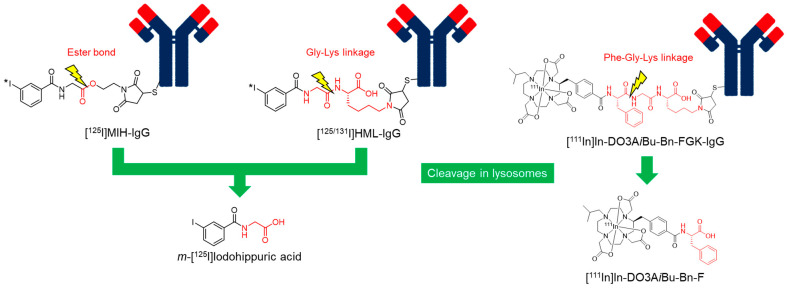
Radiolabeled antibodies with a cleavable linkage.

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
