# Peer review of "Approaches to Reducing Normal Tissue Radiation from Radiolabeled Antibodies"

_pharmaceuticals, 2024, doi:10.3390/ph17040508_

Round 1
Reviewer 1 Report
Comments and Suggestions for Authors
The manuscript "Approaches to reduce prolonged blood retention and nonspecific normal tissue accumulation of radiolabeled antibodies" by Dr. Suzuki et al. reviews approaches to reduce radiation doses to non-target tissue when radiolabeled antibodies are used. The review is divided into considering chemical dissociation of the radionuclide (section 2), prolonged blood retention / slow clearance (section 3), and accumulation in other tissues, especially liver and spleen (section 4). The review is well-written and also timely, given the current increase in the use of radiolabeled antibodies (and other radiopharmaceutical therapies).
MINOR ISSUES
1. The title of the review focuses on only two (prolonged blood retention and nonspecific normal tissue accumulation) of the three main parts of the paper (dissociation, slow clearance, liver and spleen accumulation), and the terms are not the same as those found in the section headlines. It is suggested to use a title that includes all three main parts, for instance "Approaches to reduce nonspecific tissue radiation from radiolabeled antibodies". (The authors should feel free to use a different title that this suggestion.)
2. In lines 66-69 is described how dissociation of radionuclides leads to unwanted redistribution to normal tissue, leading to decreased diagnostic accuracy "and a decrease in therapeutic efficacy and side effects of treatment." Here, the last part can be misunderstood. The side effects will be increased, not decreased. Accordingly, it is suggested to write "... efficacy and an increase in side effects."
3. In lines 118-119 is written: "Astatine has no stable isotopes and its half-life of the longest one is only 8.1 h. Such rarity has limited 211At research". This is a bit strangely expressed. Stable or long-lived isotopes of astatine would not have made 211At more easily available. Suggested rephrasing: "Astatine is not naturally occurring, as it has no stable isotopes. The isotope 211At is alpha-emitting with a half-life of 7.2 h. Its lack of general availability has, however, limited 211At research." Out of curiosity: Do you know why 210At is (the isotope with half-life 8.1 h) is not used? If yes, the explanation might be of interest to the reader (and to this reviewer).
4. In line 147 is written: "They proposed that cytochrome cytochrome-P450s (CYPs)". Is it on purpose that "cytochrome" is written twice?
5. In lines 269-270 is written: "223Ra is liberated from BFCs after the decay of 227Th, which are well known accumulates in the bone." Grammatically, the sentence indicates that it is thorium that accumulates in the bone. But the meaning seems to be that the daughter (223Ra) is accumulating in the bone, right? Suggested rephrasing: "The decay of 227Th produces 223Ra, which is liberated from BFCs by the recoil, and radium is well known to accumulate in the bone."
6. In line 365, "111In-EDTA" should be "[111In]In-EDTA" to follow current nomenclature guidelines (which the authors otherwise follow very prudently).
7. In many cases, readability at the end of lines could be improved by less or better hyphenation. Less: It may not always be worth the space to split a word if only a two or three letters remain on the line before the split. Better: When words are split, it will improve readability if compound words are split more "naturally", e.g. radio-labeling rather than radi-olabeling and radio-nuclide rather than radionu-clide. Most word processing programs allow the insertion of "soft hyphens" that show up only if the word is split; typically, a soft hyphen can be inserted with Ctrl-- (i.e. "control-hyphen").
Author Response
We would like to express our sincere gratitude to the reviewers for their constructive comments and kind suggestions. We corrected the manuscript according to the suggestions, and we believe that the revised manuscript becomes much clearer and better to read.
Replies to the comments by reviewer 1
- The title of the review focuses on only two (prolonged blood retention and nonspecific normal tissue accumulation) of the three main parts of the paper (dissociation, slow clearance, liver and spleen accumulation), and the terms are not the same as those found in the section headlines. It is suggested to use a title that includes all three main parts, for instance "Approaches to reduce nonspecific tissue radiation from radiolabeled antibodies". (The authors should feel free to use a different title that this suggestion.)
Reply: We appreciate the comment of the reviewer. We changed the title.
- In lines 66-69 is described how dissociation of radionuclides leads to unwanted redistribution to normal tissue, leading to decreased diagnostic accuracy "and a decrease in therapeutic efficacy and side effects of treatment." Here, the last part can be misunderstood. The side effects will be increased, not decreased. Accordingly, it is suggested to write "... efficacy and an increase in side effects."
Reply: We appreciate the kind comment of the reviewer. We revised the sentence as suggested.
- In lines 118-119 is written: "Astatine has no stable isotopes and its half-life of the longest one is only 8.1 h. Such rarity has limited 211At research". This is a bit strangely expressed. Stable or long-lived isotopes of astatine would not have made 211At more easily available. Suggested rephrasing: "Astatine is not naturally occurring, as it has no stable isotopes. The isotope 211At is alpha-emitting with a half-life of 7.2 h. Its lack of general availability has, however, limited 211At research." Out of curiosity: Do you know why 210At is (the isotope with half-life 8.1 h) is not used? If yes, the explanation might be of interest to the reader (and to this reviewer).
Reply: We appreciate the kind comment of the reviewer. We revised the sentences and added the reason why 210At was unsuitable for targeted alpha therapy.
- In line 147 is written: "They proposed that cytochrome cytochrome-P450s (CYPs)". Is it on purpose that "cytochrome" is written twice?
Reply: We apologize for our mistake. We corrected it.
- In lines 269-270 is written: "223Ra is liberated from BFCs after the decay of 227Th, which are well known accumulates in the bone." Grammatically, the sentence indicates that it is thorium that accumulates in the bone. But the meaning seems to be that the daughter (223Ra) is accumulating in the bone, right? Suggested rephrasing: "The decay of 227Th produces 223Ra, which is liberated from BFCs by the recoil, and radium is well known to accumulate in the bone."
Reply: We appreciate the kind comment of the reviewer. As pointed out the reviewer, we aimed to explain that the daughter radionuclide (223Ra) accumulates in the bone. We revised the sentence as suggested.
- In line 365, "111In-EDTA" should be "[111In]In-EDTA" to follow current nomenclature guidelines (which the authors otherwise follow very prudently).
Reply: We apologize for our mistake. We corrected it.
- In many cases, readability at the end of lines could be improved by less or better hyphenation. Less: It may not always be worth the space to split a word if only a two or three letters remain on the line before the split. Better: When words are split, it will improve readability if compound words are split more "naturally", e.g. radio-labeling rather than radi-olabeling and radio-nuclide rather than radionu-clide. Most word processing programs allow the insertion of "soft hyphens" that show up only if the word is split; typically, a soft hyphen can be inserted with Ctrl-- (i.e. "control-hyphen").
Reply: We apologize for the low readability of our initial manuscript. We cancelled the hyphenation settings in the revised manuscript. We also changed “pre-targeting” to “pretargeting”.
Reviewer 2 Report
Comments and Suggestions for Authors
Comments have been uploaded as attachment.

Author Response
We would like to express our sincere gratitude to the reviewers for their constructive comments and kind suggestions. We corrected the manuscript according to the suggestions, and we believe that the revised manuscript becomes much clearer and better to read.
Replies to the comments by reviewer 2
Page 13, line 534-537, “Although radiolabeled antibodies succeeded in releasing radiometabo-lites from the lysosomes, the liberated radiometabolites were retained in the lysosomes.” What does this sentence mean, kindly clarify.
Reply: We apologize for confusing the reviewer. We revised sentences as follows; “The control of releasing the designed radiometabolites in lysosomes is relatively easy. However, it is difficult to design radiometabolites that can rapidly escape from lysosomes owing to the high hydrophilicity and bulkiness of the radiometal-chelator complex. As a result, the liberated radiometabolites are often retained in the lysosomes even if the metabolizable linkage is cleaved as expected.”
Comments 1. Page 2, line 86, with a limited half-life (4.2 d) should be changed to relatively longer half-life.
Reply: We appreciate the kind comment of the reviewer. We corrected the description as pointed out.
Comments 2. Page 3, line 109, Kindly re-write the following sentence “Radiobromine shows evenly nonspecific distributed in the extracellular space” to better clarification.
Reply: We appreciate the comment of the reviewer. We changed the sentence as follows; “Radiobromine shows nonspecific accumulation in the extracellular space.”
Comments 3-5. Page 4, line 147, cytochrome word has occurred twice in the following sentence “proposed that cytochrome cytochrome-P450s (CYPs)
Page 5, line 188, Kindly correct radiometer in the following sentence “for producing radiometer -labeled anti-..”
Page 5, line 205, Instead of “BFC of Zevalin..” kindly write BFC in Zevalin”
Reply: We apologize for our mistakes. We corrected them.
Comments 6. Page 7, line 270, Kindly re-write “which are well known accumulates in the bone.”
Reply: We appreciate the comment of the reviewer. We revised the sentences as follows; “The decay of 227Th produces 223Ra, which is liberated from BFCs by the recoil, and radium is well known to accumulate in the bone.”
Comments 7. Page 8, line 319, Kindly re-write this subsection (line number 319- 324) as the concept of pre-targeting needs to be described in a better way here.
Reply: We appreciate the comment of the reviewer. We revised this section.
Comments 8-10. Page 8, line 340, thus, ..remove comma here
Page 9, line 386, were well tolerated..change it to ‘was well tolerated..’
Page 10, line 420, its rapid blood half-life.. change to..its low half-life in blood.
Reply: We appreciate the kind comments of the reviewer. We corrected them.
Reviewer 3 Report
Comments and Suggestions for Authors
The work is clear well-organized and even easy to read.
I just have three little notes that I signed in the PDF file in the attachment.
The only suggestion for improvement would be the articulation of Figure 4 with the text. In session 3. you indicate the figure, but within the paragraphs, you never link the description with the different parts (A), (B), and (C). I think it could be more effective to help the reading to have three figures instead.

The work is well-written, but some sentences could be more direct.
Author Response
We would like to express our sincere gratitude to the reviewers for their constructive comments and kind suggestions. We corrected the manuscript according to the suggestions, and we believe that the revised manuscript becomes much clearer and better to read.
Replies to the comments by reviewer 3
I just have three little notes that I signed in the PDF file in the attachment.
Reply: We appreciate the kind comments of the reviewer. We corrected them.
The only suggestion for improvement would be the articulation of Figure 4 with the text. In session 3. you indicate the figure, but within the paragraphs, you never link the description with the different parts (A), (B), and (C). I think it could be more effective to help the reading to have three figures instead.
Reply: We appreciate the comment of the reviewer. We added the citations of Figures 4A-C in the text.